# Disease risk scores for skin cancers

Pierre Fontanillas [1✉], Babak Alipanahi [1], Nicholas A. Furlotte[1], Michaela Johnson[1], Catherine H. Wilson[1] & 23andMe Research Team*, Steven J. Pitts [1], Robert Gentleman[1] & Adam Auton[1]

We trained and validated risk prediction models for the three major types of skin cancer—basal cell carcinoma (BCC), squamous cell carcinoma (SCC), and melanoma—on a cross-sectional and longitudinal dataset of 210,000 consented research participants who responded to an online survey covering personal and family history of skin cancer, skin susceptibility, and UV exposure. We developed a primary disease risk score (DRS) that combined all 32 identified genetic and non-genetic risk factors. Top percentile DRS was associated with an up to 13-fold increase (odds ratio per standard deviation increase >2.5) in the risk of developing skin cancer relative to the middle DRS percentile. To derive lifetime risk trajectories for the three skin cancers, we developed a second and age independent disease score, called DRSA. Using incident cases, we demonstrated that DRSA could be used in early detection programs for identifying high risk asymptotic individuals, and predicting when they are likely to develop skin cancer. High DRSA scores were not only associated with earlier disease diagnosis (by up to 14 years), but also with more severe and recurrent forms of skin cancer.

[1] 23andMe Inc., 223N Mathilda Ave, Sunnyvale, CA 94086, USA. *A full list of members and their affiliation present at the end of the paper.
✉email: pfontanillas@23andme.com

Cancer screening and early diagnosis are recognized as key public health strategies for reducing cancer burden. Each year, more than 14 million people worldwide are diagnosed with cancer, and 8.8 million will die from cancer, representing one in six deaths globally[1]. It has been estimated that 50–60% of cancers could be prevented or successfully treated by efficient cancer prevention and early detection programs[2,3]. Skin cancer is the most commonly diagnosed cancer, and it is also among the more preventable forms of cancer. In the United States, one in five Americans are likely to develop skin cancer during their lifetime[4]. Even melanoma, the more aggressive form among the three main skin cancers, which include basal cell carcinoma (BCC) and squamous cell carcinoma (SCC), has a 5-year survival rate of 98%, if diagnosed at an early stage[5]. Despite being preventable, more than 15,000 Americans die every year from skin cancer[5,6].

Public health efforts for prevention have focused on education regarding the danger of sun exposure and increasing adoption of sun-protective habits[7]. For early skin cancer detection, medical organizations have recommended full body skin examination, a quick, inexpensive, and noninvasive method. Unfortunately, there is no convincing evidence that visual skin examination is an effective method to detect skin cancer in the general population[8]. Furthermore, epidemiological studies have identified many additional risk factors, including skin pigmentation, skin susceptibility to sun exposure, family history of skin cancer, and genetics[9]. There is thus a considerable interest in the development of new cost-effective screening programs that integrate all risk factors, environmental and genetics, and accurately identify asymptomatic individuals with high risk of developing skin cancer[10]. The ultimate goals of screening would be to identify individuals at high risk, construct personalized skin cancer monitoring plans for these individuals, detail how early and frequently skin cancer detection programs should be implemented, and to anticipate the type and intensity of potential treatments.

Many skin cancer predictive models have been proposed over the years[9,11]. Unfortunately, they have generally suffered from insufficient validation, an incomplete catalog of known risk factors, including genetics, and relied heavily on the current age of individuals, which limits their application for early detection. Building medically relevant individual risk scores is challenging, but the recent emergence of polygenic risk scores (PRS) has provided a potential blueprint for success[12,13]. The growth of interest in PRS has been fueled by several major innovations: the development of prediction models in large training datasets, the recognition that while risk scores can be weak predictors overall, their long tails permit the identification of individuals with high disease risk, and finally the ability to validate and characterize risk score distributions in large, independent, and preferably longitudinal cohorts. The open questions are whether non-genetic risk scores would also exhibit long tail behavior, and if they can be combined to improve individual risk predictions and their clinical utility[14].

In this study, we deployed an online cancer survey, which contained questions about personal and family histories of the three main skin cancers (BCC, SCC, and melanoma), as well as risk factors identified after reviewing the recent skin cancer literature[9,11]. Over the course of approximately thirteen months from May 2016 to June 2017, more than 210,000 research participants responded to questions from an in-house designed cancer survey. (Fig. 1a, and Supplementary Tables 1, 2). The baseline survey contained 34 questions regarding personal history of skin cancer (including skin cancer type, age at diagnosis, body location, prescribed treatments, and information regarding cancer recurrence), 12 questions regarding the family history of skin cancer (skin cancer type for close relatives, including parents, siblings, and children), and 23 questions regarding risk factors and exposures (including skin, hair, and eye pigmentation, freckles, moles, skin sensitivity to sun, tanning, sunburns, and sun/UV exposure). Using data from 103,008 participants collected during the first four months after survey deployment, we selected and trained predictive models for each of the three skin cancers independently. The resulting models contained a total of 32 risk factors, including 20 factors from the cancer survey, 11 factors identified from the wider 23andMe database, and a PRS. For interpretability, we grouped the 31 non-genetic risk parameters into six separate risk scores. Finally, we defined two global disease risk scores, which we refer to as DRS and DRSA, respectively. The DRS included all 32 risk factors, while the DRSA excluded age effects. The DRS provided a global risk score of developing skin cancer. The DRSA, which we demonstrated to be largely age independent, permitted to study disease trajectories, and to predict incidence rates and age at onset. The performance of all risk scores was assessed in a validation set consisting of 88,924 participants. In addition of the baseline survey, we also asked all participants in the validation set to complete a follow-up survey in both 2018 and 2019 and provide updated information regarding their skin cancer disease status and treatments during the preceding 12 months. We used the responses from 49,501 participants to quantify incident cases and cancer free participants, in groups of individuals with different predicted risks of skin cancer.

## Results

**Identifying skin cancer risk factors.** The demographic characteristics of the training and validation sets are described in Supplementary Table 1, and the participants' geographical provenance is shown in Supplementary Fig. 1. We restricted analyses to participants with current age (at the time of the baseline survey) between 30 and 90 years. The sex ratio of participants was biased toward females (~1.5x females), and close to 50% of participants reported a current age between 50 and 70 years. The age distribution and sex ratio were in line with the general characteristics of the 23andMe research cohort. The training set used to select factors (including the variants for PRS) and to train the predictive models contained 14,898 BCC, 7479 SCC, and 3998 melanoma cases (Fig. 1b).

We identified 32 risk factors that contributed to at least one type of skin cancer, following the procedure described in Fig. 1c (see "Methods" for a detailed description). As the three skin cancers share common risk factors, and in order to enable comparisons, we included all 32 identified factors in each skin cancer final model. The 32-factor models explained 21.6%, 20.0%, and 19.8% of phenotypic variance of BCC, SCC, and melanoma, respectively (Fig. 2 and Supplementary Table 9). The following section describes the main risk factors included in these models, and their contribution to each skin cancer risk. We separated factors commonly used in skin cancer prediction models[9,15] from factors that are generally not included (although not necessarily novel).

*Commonly used risk factors.* Genetic variants included in PRS were directly selected in the training set using a simple clumping and thresholding method (Supplementary Table 10 and Supplementary Fig. 5). The BCC, SCC, and melanoma PRS included 47, 14, and 18 variants, respectively, and explained between 1.5% and 3% of the variance in skin cancer risk. While many GWAS on skin cancers have been published recently, we chose to restrict the PRS to only contain variants discovered within the training set. The majority of recently published skin cancer GWAS include 23andMe data in their analyses, and are, therefore, not

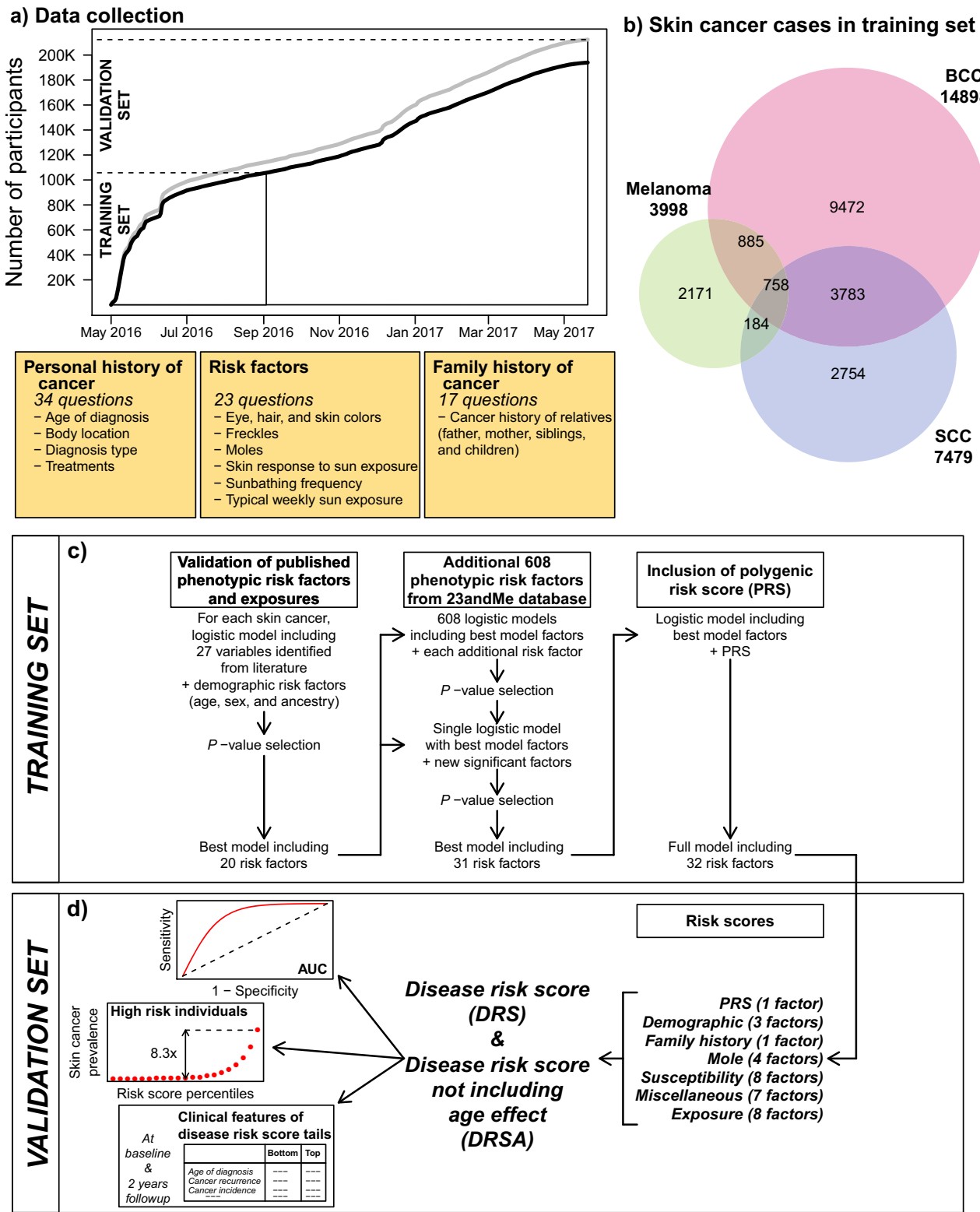

**Fig. 1 Data collection for the training and validation sets, description of the cancer survey, overview of the predictive model construction, and disease risk scores. a** Cumulative data collection from May 2016 to September 2016. Participants recruited between May and September 2016 were included in the training set. The gray line is the total number of participants from European ancestry. BCC, SCC, and melanoma prediction models were trained on participants 30–90 years old (black line). **b** Number of skin cancer cases in the training set. **c** Overview of prediction model construction in the training set. **d** Disease risk scores and predictive performances evaluated in the validation set.

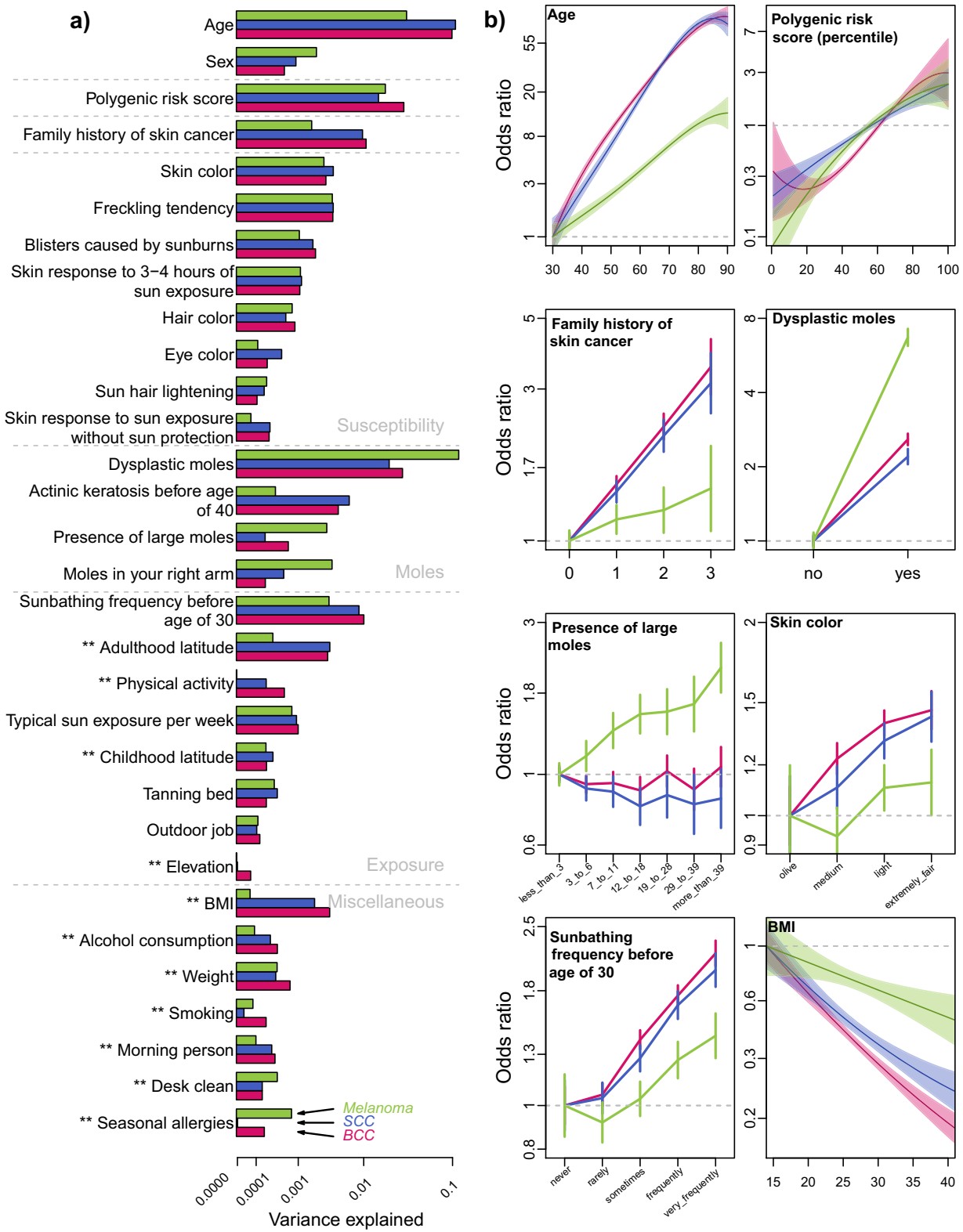

**Fig. 2 Variance explained and risk factor effects in the final 32-factor models.** The training set included 103,008 participants (14,898 BCC, 7479 SCC, and 3998 melanoma cases). **a** Risk factors are organized and presented by risk scores. The 5 ancestry PCs are not shown. The variance explained is the deviance of the model, a standard measure of goodness of fit of the model that approximates the variance explained. Deviances were recalculated in the full models, after ranking the risk factors in each skin cancer, based on the deviance explained by each individual risk factor, from the larger to the smaller deviances. ** Indicates risk factors that were not included in the cancer survey, and were identified from the wider 23andMe database. **b** A sample of risk factor effects (estimated effects and 95% CI). Continuous risk factors, age, and BMI, were modeled as polynomial variables.

independent of the PRS from this study[16,17]. As such, including associations from other publications could bias the assessment of our risk models in the validation set. An exception to this issue is provided by a recent large-scale PRS study of melanoma, which we were able to use to compare to our melanoma PRS[18]. The results are summarized in Supplementary Fig. 14, and showed that the external melanoma PRS, which combined 204 variants, achieved a slightly better area under the curve (AUC) than the melanoma PRS from the present study (0.594 vs 0.579). The studies also showed an excellent correlation between effect estimates, in particular for 12 variants with a reasonable association confidence (P-value $< 1.0e^{-3}$) in the training set (correlation $r = 0.92$; for the remaining 192 variants, $r = 0.59$).

Only two risk factors—current age and the presence of dysplastic moles—surpassed the explanatory performance of the PRS. Age was strongly predictive of BCC and SCC ($r^2 = 10\%$) but less so of melanoma ($r^2 = 3.3\%$). The presence of dysplastic moles increased the relative risk of melanoma by 6.7×, and explained 11.5% of the melanoma variance, but was less predictive for BCC and SCC, explaining 2.9% and 2.1% of the variance, respectively. In addition to dysplastic moles, the number and size of moles were generally more predictive of melanoma than BCC and SCC. On the other hand, a diagnosis with actinic keratosis before age 40 increased BCC and SCC risks by >3×. A family history risk score, computed as a simple sum of binary indicators of reported skin cancer in father, mother, siblings, and children, explained about 1% of the BCC and SCC variance, but only 0.3% of the melanoma variance. Pigmentation factors individually explained less than 1% of skin cancer variance, with consistent estimates between the three skin cancers. Among the exposure risk factors, sunbathing frequency before age 30 explained between 0.3% and 1% of the skin cancer variance. The typical sun exposure per week, which was measured by the current reported number of minutes per week spent in the sun, also increased the risk of skin cancer, and explained about 0.1% of variance. Tanning bed lifetime usage, which was reported by 41.6% of the participants, was associated with a moderate increase of skin cancer risk (maximum 1.3× in SCC for participants that reported using tanning beds more than 30 times). Finally, sex showed a moderate and consistent effect across the three skin cancers, with men having ~1.6x higher risk of developing skin cancer than women.

*Additional risk factors.* Living at low latitudes and high elevation during both childhood and adult life moderately increased the risk of skin cancer, in particular for BCC and SCC (1.3×), and explained up to 0.3% of the variance[19,20]. More surprisingly, both BMI and weight were retained in the final set of risk factors of each skin cancer, and showed opposite effects: larger weight increased the risk of skin cancer whereas a larger BMI decreased it (see "Methods" for collinearity analysis). The decreased risk with high BMI was large, with a 4× less risk of BCC cancer for participants with a BMI > 35, but it was only half of this effect in melanoma. It has been suggested that a direct protective link between obesity and skin cancer is unlikely, and may instead reflect other life-style factors[21,22]. Although the weight effects were more modest, with a 2.2 to 3.1× increase for participants reporting a current weight >300 lb relative to participants with a weight ranging between 100 and 200 lb, they could indicate direct effects of physiological and physical (increase of skin surface) changes in overweight participants[23]. A few other risk factors included in the final 32-factor models could also be surrogates of sun exposure behavior. For example, participants self-reporting a morning person chronotype showed a small increase of skin cancer risk (1.05 to 1.1×) relative to night person chronotype. Participants reporting as having engaged in intense physical activities also showed a slight (1.1×) increase of skin cancer risk.

Curiously, a so-called 'clean desk' factor (referring to a participant's self-reported preference for keeping their desk clean) was selected as the best representative variable from a cluster of correlated personality traits (see "Methods" and Supplementary Fig. 4), and showed a small inverse correlation with skin cancer risk. Finally, participants that reported as having ever smoked showed a slight reduction (1.06×) of BCC risk. While there have been sporadic reports of smoking being protective for skin cancer[24,25], it appears more likely that smoking status could be acting as a proxy for behaviors associated with smoking and not directly reflect smoking effects.

**Grouping risk factors into risk scores.** For descriptive convenience, we grouped the non-genetic risk factors into six risk scores on the basis of physiological or behavioral similarity (Fig. 1d and "Methods"). The 'Demographic risk score' includes three factors; age, sex, and genetic ancestry. The 'Family history risk score' contains only the family history risk factor. The 'Mole risk score' combines four risk factors related to the presence or frequency of moles (dysplastic moles, presence of large moles, number of moles on the right arm), or skin condition (diagnosis with actinic keratosis before the age of 40). The 'Susceptibility risk score' combines 8 factors related to pigmentation and skin reaction to sun exposure (skin, eye, and hair colors, number of freckles on face and body, number of blisters caused by sunburns, and sun hair lightening). The 'Exposure risk score' combines 8 factors estimating the lifetime or current sun exposure (sunbathing frequency before age 30, tanning bed usage, childhood and adulthood latitude and elevation, typical sun exposure per week, outdoor job, and physical activity). We combined the 7 remaining factors into a 'Miscellaneous risk score' (BMI, weight, smoking, alcohol consumption, seasonal allergies, morning, and desk clean person). Finally, we constructed two main disease risk scores, called DRS and DRSA. DRS includes all 32 risk factors identified in the factor selection process, whereas DRSA excludes age effects. Risk scores were calculated for each participant as the sum of the factor responses weighted by the effect size obtained by the 32-parameter model in the training set (Supplementary Tables 10, 14).

**Risk score prediction performance and correlations.** We evaluated the predictive performances of each risk score in the validation set. We computed the receiver operating characteristic (ROC) and the precision-recall (PR) curves, and estimated the AUC (Supplementary Fig. 6). We also plotted the prevalence of the three skin cancers across risk score distributions binned into percentiles, and estimated the change in prevalence between the top percentile bin and the middle of risk score distribution. With almost 89,000 participants in the validation set, the prevalence in each bin was computed on the basis of 890 participants (Fig. 3 and Supplementary Fig. 7). In the following section, we focus on the performance of DRS, DRSA, and the PRS alone.

*DRS.* This score combined all 32 risk factors, and obtained good predictive performance across all three skin cancers, with an AUC = 0.79, 0.80, and 0.78 for BCC, SCC, and melanoma, respectively. This corresponds to an odds ratio per standard deviation increase of 3.60 (95% CI: 3.51–3.69), 3.52 (3.42–3.63), and 2.58 (2.52–2.66), respectively. The upper tail of the DRS distribution showed substantial increase risk of developing skin cancer: the prevalence of skin cancer for participants in the top DRS percentile was 69.8%, 46,7%, and 31.8% for BCC, SCC, and melanoma, respectively. These correspond to a 5.2×, 8.1×, and 12.9× risk increase relative to participants with middle DRS percentile (Fig. 3). Although DRS showed excellent ability to

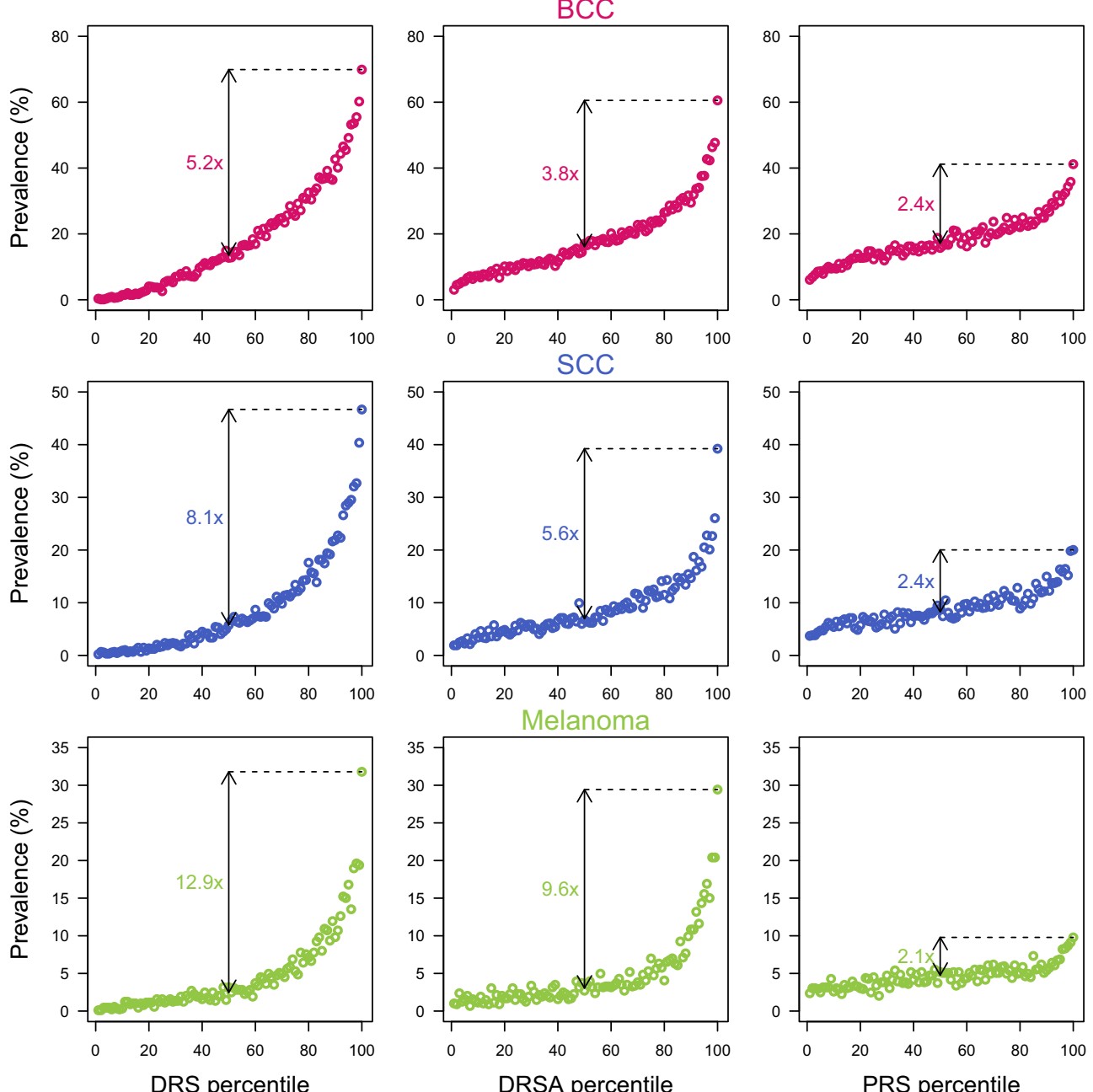

**Fig. 3 Prevalence of skin cancer cases across the binned DRS, DRSA, and PRS distributions in the validation set.** Risk score distributions are binned into percentiles. Each bin contains 890 participants. The prevalence is the percentage of participants reporting skin cancer in each bin.

identify high risk individuals for the three skin cancers, they are heavily driven by current age (Supplementary Fig. 8), and hence it limits their application in early detection programs.

*DRSA.* In contrast to the DRS, the DRSA are not including the age effects. They however use the same weights for the remaining 31 factors than the ones used for the DRS calculations. These weights were, by definition, adjusted for age by the linear modeling (Supplementary Table 9). Since many risk factors showed age dependency (Supplementary Fig. 16), and because the linear modeling could have produced incomplete age adjustment, we deployed a set of analyses and demonstrated that the DRSA were largely independent of age in the three skin cancers (Supplementary Figs. 8, 17, 19). As a consequence, within skin cancer, DRS and DRSA were only moderately correlated with each other,

with $r = 0.64$, 0.56, and 0.74 in BCC, SCC, and melanoma, respectively (Supplementary Table 11). DRSA achieved lower overall prediction performances than DRS (AUC = 0.69, 0.68, and 0.73, for BCC, SCC, and melanoma, respectively), which was expected given the age contribution, but nonetheless retained strong risk enrichment in upper tails, with 3.7x, 5.3x, and 9.8x increase risk for participants in the top percentile for BCC, SCC, and melanoma respectively, and which corresponded to diagnosis rates of 60.1%, 38.6%, and 28,7% (Fig. 3). Across the full DRSA distributions, the odds ratios per standard deviation of DRSA increase were 1.98 (1.95–2.02), 1.83 (1.79–1.87), and 2.08 (2.03–2.13), respectively. We also explored the DRSA correlation between skin cancers. Although the non-melanoma DRSA were highly correlated with each other ($r = 0.91$), a principal component analysis showed different contributions of the Susceptibility

and Exposure risk scores that enable the identification of different high risk participants for BCC and SCC (Supplementary Fig. 9). Despite sharing of the included risk factors, the melanoma DRSA showed only a moderate correlation with non-melanoma DRSA ($r = 0.74$ for both BCC and SCC comparisons).

*PRS.* As expected, PRS were independent of current participant age (Supplementary Fig. 8), and showed modest predictive performances with AUC = 0.62, 0.60, and 0.58 for BCC, SCC, and melanoma, respectively. This corresponds to a odds ratio per standard deviation increase for PRS of 1.57 (1.55–1.60), 1.44 (1.41–1.48), and 1.33 (1.29–1.37), respectively. For participants in the top PRS percentile, about 41.1%, 20.0%, and 9.9% reported BCC, SCC, and melanoma, respectively, corresponding to a 2.1 to 2.4 fold increased risk compared to participants with an average PRS (Fig. 3).

All risk scores generally showed long tail behaviors (Supplementary Fig. 7). Beside the Demographic risk scores, which exhibited a 2.6 to 4 fold enrichment in the tails, the Mole risk score achieved good predictive performances for melanoma, with an AUC = 0.67 and a top percentile with a 6.9 fold increase risk of developing melanoma. Interestingly, the tail of the distribution with elevated risk was unusually large: the top 15% of participants with a high Mole risk score had >3 fold risk of developing melanoma. Similarly, despite AUC < 0.59, the BCC and SCC Mole risk scores showed large risk increases for the top percentile (2.4 and 3.9 folds, respectively). Conversely, Exposure risk scores generally showed poor predictive performance with AUC ~0.55, a top percentile increased risk of <1.5 fold, and no clear evidence that the upper tail had strong disease enrichment. Interestingly, these individual risk scores were also independent of age, and had a constant contribution to DRSA scores across age (Supplementary Fig. 19).

**Medical utility of DRS, DRSA, and PRS.** We evaluated the putative medical utility of the skin cancer risk scores by first characterizing disease features for participants reporting skin cancer within the validation set, at baseline. We first looked at the age of diagnosis of skin cancer, which was not used during the development of the predictive models. The age independent DRSA were remarkably powerful at predicting early age of diagnosis in all three skin cancers: participants in the top percentile were diagnosed on average 10–14 years earlier than the participants with average scores (Fig. 4). High PRS, Mole, Susceptibility, and Exposure risk scores all predicted early ages of diagnosis as well (Supplementary Fig. 10). Conversely, because of their strong age dependencies, DRS were poor predictors of age of diagnosis. We then analyzed other characteristics of skin cancers at baseline and we also observed that high DRS, DRSA, and PRS all predicted a higher past recurrence rate of skin cancer, and a higher probability of developing multiple types of skin cancer (Table 1).

Using the prospective data collection (with data collected on an annual basis for two years after the baseline survey), we investigated the ability of the DRS, DRSA, and PRS to predict incident cases of skin cancer, and their age of diagnosis. We defined incident cases as participants who did not report to have been diagnosed with skin cancer in the original baseline survey, but subsequently reported a diagnosed for skin cancer during the previous year in the follow up survey. We further defined cancer free participants as participants who reported to have been diagnosed with skin cancer in the original baseline survey, but who also reported not been treated for skin cancer during the previous year in the follow-up survey. Using these metrics, we showed that high baseline DRS, DRSA, and PRS were all

associated with a higher rate of incident cases in the prospective data, and fewer cancer free participants overall. Consistent with the results from the baseline survey, we also showed that non-cancer participants with high baseline DRS, DRSA, and PRS developed new skin cancer earlier than participants with lower risk scores (Table 1, and Supplementary Figs. 11, 12). Finally, building on the age-independence property of the DRSA and combining with the skin cancer prevalences observed in the validation set, we plotted the lifetime risk trajectories for the three skin cancers (Fig. 5). From these trajectories, we derived the expected incidence rates and ages of diagnosis (see Methods), and compared them to the observed values in the validation and longitudinal dataset. We observed an excellent concordance between the expected and observed 2-years incidence rates (Supplementary Fig. 18) and the ages of diagnosis, in particular for individuals with high risk of developing skin cancer (Fig. 5).

**Discussion**
In this study, we have shown that individual risk scores can predict relevant medical outcomes for skin cancer and offer the potential for early detection applications. These scores utilize the long tail behavior of the genetic and non-genetic factors, and potentially enable the identification of asymptomatic individuals with elevated skin cancer risk. We also showed that these powerful risk scores can be simply constructed by combining risk factors in an additive fashion. While we explored more complex models, including potential interaction between risk factors[26], we did not find them to significantly improve the prediction performance (Supplementary Tables 12, 13, and Supplementary Fig. 13).

Our risk models were built using self-reported skin cancer diagnoses, and multiple studies have documented variable rates of self-reporting accuracy in skin cancers[27,28]. Misclassification arising from self-reported information could lead to the development of suboptimal prediction models[29]. While we do not have the necessary clinical data to directly estimate a misclassification rate for the three skin cancers, we were able to indirectly estimate the rate misclassification for melanoma by focusing on the genetic variant effect estimates obtained in our study, and comparing to those obtained by a study based on participants diagnosed with pathology or histopathology-confirmed invasive cutaneous melanoma[18]. This analysis, summarized in Supplementary Fig. 14, showed a high correlation between the effect size estimates ($r = 0.92$ [0.76–0.97]) and no evidence of dilution between variant effect estimates from the training set and the independent melanoma GWAS, suggesting a comparable level of misclassification in the 23andMe cohort and the independent melanoma cohort. Similar analyses for BCC and SCC are complicated for the fact that most recent large GWAS publications of these diseases have included 23andMe data, and the variant effect estimates are therefore not independent. However, for BCC, a study[17] directly estimated self-reported misclassification in 23andMe cohort, with adjudicated medical records, and revealed a sensitivity and specificity of 93% and 99%, respectively.

Age remains a key component of the skin cancer predictive models. It not only captures the detrimental effects of senescence processes, but it is also correlated with many of the non-genetic risk factors. Nevertheless, we demonstrated that DRSA, which combined all identified risk factors, except age, was an age independent predictor of skin cancer for 30 years or older individuals. This age independence allowed to tread the DRSA and age as two independent variables, and it permitted to draw the lifetime risk trajectories, stratified by DRSA scores, for the three skin cancers (Fig. 5). In return, the trajectories permitted to derive the expected incidence rate and age of diagnosis of skin cancers

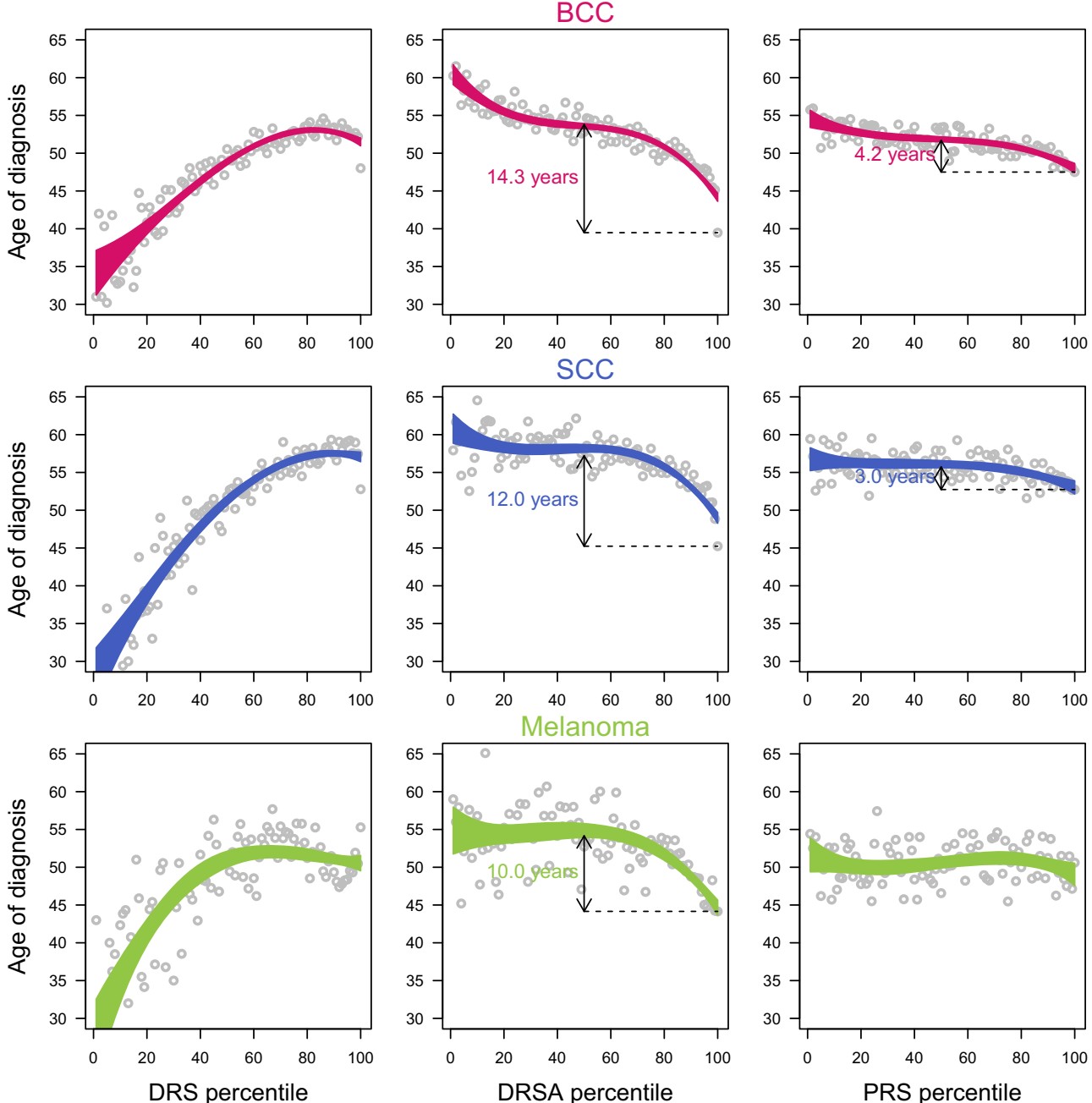

**Fig. 4 Mean age of skin cancer diagnosis across the DRS, DRSA, and PRS distributions in the validation set.** Risk score distributions are binned into percentiles. Each bin contains 890 participants. The dots are the mean age of diagnosis in each bin, and the area represents the 95% confidence interval of the mean age of diagnosis.

across the DRSA distributions. By comparing these expected values to the 2-years incidence rates and ages of diagnosis observed in the follow-up cohort, we concluded that the lifetime risks, which were obtained from the prevalences in the validation set, were well-calibrated for individuals with high risk of developing skin cancer (i.e. with a DRSA score in the top decile). Altogether, it indicates that the DRSA and the lifetime risk trajectories can be used in early detection programs to identify asymptotic individuals with high risk of developing skin cancer, and to predict when they are likely to develop the disease.

There remains considerable scope for further improvement. The current risk scores could be refined by a better characterization of the genetics and of the individual risk behaviors

associated to sun exposure. In order to enable a valid test of our models, we directly selected PRS variants in the training set, which had a limited statistical power for detecting associations. Larger skin cancer GWAS would expand the set of associated variants for each skin cancer, and potentially improve the prediction performance of the PRS, as would the inclusion of known pathogenic variants for skin cancers not identified by GWAS[30–32]. Additional work is also needed to understand the prediction performance in the different European sub-populations, and the portability to non-European ancestries. The current BCC and SCC models seem to perform equally well for individuals of Northern, Southern, or Eastern European ancestry, but the melanoma model shows signs of performance degradation

**Table 1 Disease characteristics of the bottom, middle (47.5 to 52.5 percentiles), and top 5% of the DRS, DRSA, and PRS distributions in the validation set for each three skin cancer.**

| | BCC | | | SCC | | | Melanoma | | |
|---|---|---|---|---|---|---|---|---|---|
| | **Bottom** | **Middle** | **Top** | **Bottom** | **Middle** | **Top** | **Bottom** | **Middle** | **Top** |
| **DRS** | | | | | | | | | |
| At baseline: | | | | | | | | | |
| No. of cases | 14 | 604 | 2593 | 19 | 275 | 1608 | 14 | 116 | 916 |
| Sex (% of females) | 85.7 | 66.2 | 48.5 | 78.9 | 65.1 | 45.1 | 78.6 | 66.4 | 46.1 |
| Age | 40.4 (7.2) | 60.7 (9.8) | 72.0 (8.8) | 36.1 (6.4) | 60.0 (8.3) | 73.3 (8.5) | 40.4 (8.3) | 62.2 (10.8) | 66.2 (10.3) |
| Age of diagnosis | 33.7 (8.8) | 49.0 (12.1) | 51.4 (14.5) | 26.7 (6.6) | 51.1 (11.5) | 56.8 (13.7) | 26.9 (9.2) | 51.5 (14.1) | 51.9 (13.9) |
| Cancer stage at diagnosis | 0.00 (0.00) | 1.06 (0.24) | 1.24 (0.51) | 0.00 (0.00) | 1.08 (0.28) | 1.27 (0.53) | 0.00 (0.00) | 1.33 (0.58) | 1.33 (0.53) |
| No. cancer diagnosed in the last two years | 1.36 (0.50) | 1.57 (0.66) | 1.97 (0.89) | 1.26 (0.56) | 1.56 (0.60) | 1.83 (0.81) | 1.21 (0.43) | 1.41 (0.64) | 1.36 (0.60) |
| No. of different type of skin cancer | 1.07 (0.27) | 1.22 (0.45) | 1.69 (0.67) | 1.05 (0.23) | 1.47 (0.56) | 2.00 (0.62) | 1.07 (0.27) | 1.41 (0.68) | 1.84 (0.83) |
| Incident cases: | | | | | | | | | |
| No. of incident cases | <5 | 57 | 123 | <5 | 34 | 113 | <5 | 18 | 36 |
| Sex (% of females) | na | 66.7 | 44.7 | na | 64.7 | 35.4 | na | 72.2 | 44.4 |
| Age | na | 61.6 (9.1) | 71.8 (8.7) | na | 59.0 (7.2) | 73.6 (7.8) | na | 59.1 (10.2) | 67.5 (10.8) |
| % of incident cases | na | 2.62 | 11.98 | na | 1.46 | 7.13 | na | 0.73 | 1.72 |
| Cancer free: | | | | | | | | | |
| No. of cancer free | 5 | 277 | 928 | 6 | 139 | 614 | 8 | 54 | 463 |
| Sex (% of females) | 50 | 66.7 | 44.7 | 100 | 64.7 | 35.4 | 100 | 72.2 | 44.4 |
| Age | 44.5 (19.1) | 61.6 (9.1) | 71.8 (8.7) | 41.0 (0.0) | 59.0 (7.2) | 73.6 (7.8) | 36.0 (0.0) | 59.1 (10.2) | 67.5 (10.8) |
| % of cancer free | 100 | 84.7 | 59.6 | 100 | 86.9 | 64.5 | 100 | 91.5 | 90.1 |
| **DRSA** | | | | | | | | | |
| At baseline: | | | | | | | | | |
| No. of cases | 208 | 715 | 2125 | 100 | 299 | 1160 | 63 | 148 | 906 |
| Sex (% of females) | 69.2 | 59.3 | 58 | 71 | 53.5 | 53.3 | 77.8 | 45.3 | 51.5 |
| Age | 71.4 (8.9) | 66.9 (10.2) | 61.6 (10.9) | 68.2 (11.2) | 69.4 (9.7) | 63.9 (10.0) | 69.0 (9.0) | 66.7 (10.9) | 58.5 (13.2) |
| Age of diagnosis | 59.0 (13.4) | 53.6 (13.0) | 44.5 (12.7) | 58.9 (13.5) | 57.7 (12.1) | 49.2 (12.4) | 54.3 (13.7) | 52.0 (14.6) | 45.3 (14.8) |
| Cancer stage at diagnosis | 1.20 (0.42) | 1.20 (0.48) | 1.23 (0.51) | 1.00 (0.00) | 1.29 (0.61) | 1.26 (0.57) | 1.50 (0.71) | 1.14 (0.38) | 1.32 (0.54) |
| No. cancer diagnosed in the last two years | 1.41 (0.55) | 1.66 (0.72) | 1.91 (0.84) | 1.43 (0.63) | 1.68 (0.71) | 1.79 (0.77) | 1.20 (0.45) | 1.34 (0.62) | 1.36 (0.58) |
| No. of different type of skin cancer | 1.14 (0.35) | 1.30 (0.51) | 1.63 (0.66) | 1.35 (0.52) | 1.62 (0.58) | 1.98 (0.62) | 1.35 (0.60) | 1.54 (0.72) | 1.70 (0.80) |
| Incident cases: | | | | | | | | | |
| No. of incident cases | 21 | 47 | 97 | 8 | 26 | 91 | <5 | 11 | 28 |
| Sex (% of females) | 71.4 | 55.3 | 53.6 | 62.5 | 61.5 | 46.2 | na | 45.5 | 57.1 |
| Age | 71.2 (10.6) | 65.5 (9.0) | 60.2 (10.6) | 68.0 (10.8) | 68.8 (10.5) | 63.3 (10.1) | na | 62.3 (14.5) | 59.6 (12.0) |
| % of incident cases | 0.91 | 2.35 | 7.78 | 0.34 | 1.16 | 5.03 | na | 0.48 | 1.4 |
| Cancer free: | | | | | | | | | |
| No. of cancer free | 102 | 285 | 811 | 50 | 124 | 466 | 32 | 85 | 439 |
| Sex (% of females) | 71.4 | 55.3 | 53.6 | 62.5 | 61.5 | 46.2 | 100 | 45.5 | 57.1 |
| Age | 71.2 (10.6) | 65.5 (9.0) | 60.2 (10.6) | 68.0 (10.8) | 68.8 (10.5) | 63.3 (10.1) | 72.0 (0.0) | 62.3 (14.5) | 59.6 (12.0) |
| % of cancer free | 82.3 | 75.2 | 64.2 | 92.6 | 75.2 | 66.3 | 100 | 95.5 | 90.7 |
| **PRS** | | | | | | | | | |
| At baseline: | | | | | | | | | |
| No. of cases | 336 | 747 | 1559 | 182 | 398 | 776 | 123 | 215 | 392 |
| Sex (% of females) | 60.1 | 56.5 | 57.4 | 59.9 | 58.8 | 55.8 | 54.5 | 53 | 55.4 |
| Age | 67.6 (10.6) | 66.4 (10.2) | 64.4 (10.8) | 66.4 (10.1) | 67.9 (10.7) | 66.1 (9.3) | 63.1 (11.0) | 63.2 (12.1) | 62.5 (13.2) |
| Age of diagnosis | 54.0 (14.2) | 53.0 (12.9) | 48.2 (13.2) | 55.7 (13.2) | 56.3 (12.8) | 53.2 (12.4) | 51.6 (15.4) | 49.7 (14.8) | 49.1 (15.7) |
| Cancer stage at diagnosis | 1.11 (0.32) | 1.22 (0.42) | 1.24 (0.52) | 1.00 (0.00) | 1.17 (0.38) | 1.27 (0.62) | 1.33 (0.52) | 1.25 (0.45) | 1.23 (0.43) |
| No. cancer diagnosed in the last two years | 1.58 (0.70) | 1.66 (0.71) | 1.88 (0.84) | 1.44 (0.62) | 1.66 (0.74) | 1.78 (0.81) | 1.44 (0.68) | 1.39 (0.57) | 1.35 (0.61) |
| No. of different type of skin cancer | 1.30 (0.54) | 1.35 (0.55) | 1.50 (0.61) | 1.48 (0.63) | 1.74 (0.63) | 1.83 (0.62) | 1.41 (0.68) | 1.65 (0.78) | 1.76 (0.80) |
| Incident cases: | | | | | | | | | |
| No. of incident cases | 31 | 60 | 78 | 25 | 45 | 73 | 6 | 8 | 32 |
| Sex (% of females) | 51.6 | 50 | 56.4 | 56 | 46.7 | 54.8 | 66.7 | 37.5 | 46.9 |
| Age | 66.4 (12.5) | 65.2 (10.9) | 62.0 (11.2) | 66.4 (10.3) | 67.0 (10.2) | 66.1 (9.9) | 63.7 (12.8) | 65.4 (12.2) | 64.6 (13.2) |
| % of incident cases | 1.4 | 2.9 | 5 | 1.1 | 2 | 3.6 | 0.3 | 0.3 | 1.4 |
| Cancer free: | | | | | | | | | |
| No. of cancer free | 149 | 324 | 571 | 77 | 180 | 314 | 57 | 105 | 185 |
| Sex (% of females) | 51.6 | 50 | 56.4 | 56 | 46.7 | 54.8 | 66.7 | 37.5 | 46.9 |
| Age | 66.4 (12.5) | 65.2 (10.9) | 62.0 (11.2) | 66.4 (10.3) | 67.0 (10.2) | 66.1 (9.9) | 63.7 (12.8) | 65.4 (12.2) | 64.6 (13.2) |
| % of cancer free | 78.8 | 75.3 | 62.9 | 88.5 | 73.8 | 68.4 | 89.1 | 89 | 86.4 |

Mean and SD (within parenthesis) are shown for the disease characteristics. Statistics for categories with counts <5 are masked (na) to avoid risk of participant re-identification.
*No.* number.

(Supplementary Fig. 15). For the portability to non-European ancestries, it has been regularly reported that PRS developed in European cohorts underperform in non-European cohorts[33], and such issues are likely to be even more challenging for the non-genetic risk factors, because trans-ethnic data collection are generally very heterogeneous[34].

## Methods

**23andMe cohort and data collection**. All participants were drawn from the customer base of 23andMe, Inc., a consumer genetics company. Participants provided informed consent and participated in the research online, under a protocol approved by the external AAHRPP-accredited IRB, Ethical & Independent Review Services (www.eandireview.com). Over the course of approximately thirteen months from May 2016 to June 2017, more than 210,000 research participants

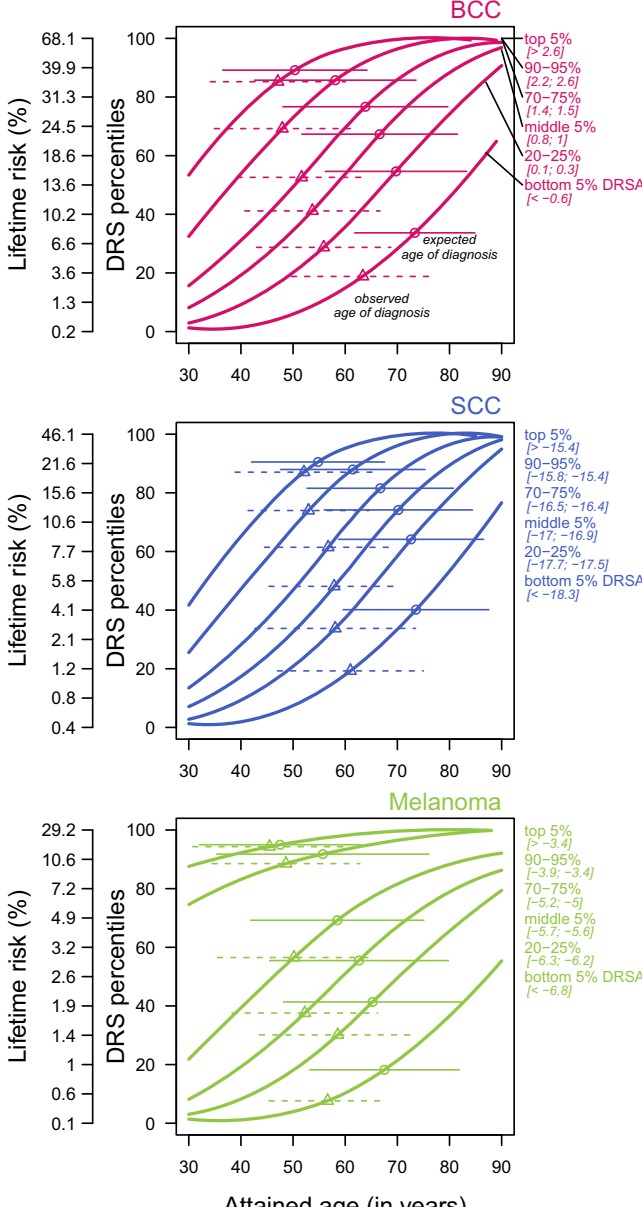

**Fig. 5 Lifetime skin cancer risk stratified by DRSA percentiles.** The expected ages of diagnosis (mean and SD) were computed using yearly incidence rates derived from the lifetime risk curves, and assuming CDC survival estimated rates, in White American, for the year 2017 (see "Methods" for details). The observed ages of diagnosis (mean and SD) were calculated in the validation set (88,924 participants, see Supplementary Table 1).

(Supplementary Table 1) responded to questions from an in-house designed cancer survey (Supplementary Table 2). The baseline survey contains 34 questions regarding personal history of skin cancer (including skin cancer type, age at diagnosis, body location, prescribed treatments, and information regarding cancer recurrence), 12 questions regarding the family history of skin cancer (skin cancer type for close relatives, including parents, siblings, and children), and 23 questions regarding risk factors and exposures (including skin, hair, and eye pigmentation, freckles, moles, skin sensitivity to sun, tanning, sunburns, and sun/UV exposure). The list of candidate risk factors and exposures has been compiled from two recent systematic reviews of thousands of epidemiological studies that have evaluated risks associated with a wide range of environmental and phenotypic factors[9,11]. Only participants who completed the full survey were included in the analysis. The geographical distribution of collected skin cancer across USA territory is shown in Supplementary Fig. 1.

Each participant received a yearly health follow-up survey in 2018 and 2019. The survey asked if the participants received treatments or were diagnosed for the three skin cancers during the last twelve months.

**Genetic data and selection of unrelated participants with European ancestry.** Samples were genotyped on one of four genotyping platforms. The V1 and V2 platforms were variants of the Illumina HumanHap550 + BeadChip, including about 25,000 custom SNPs selected by 23andMe, with a total of about 560,000 SNPs. The V3 platform was based on the Illumina OmniExpress + BeadChip, with custom content to improve the overlap with our V2 array, with a total of ~950,000 SNPs. The V4 platform is a fully custom array, including a lower redundancy subset of V2 and V3 SNPs with additional coverage of lower-frequency coding variation, and ~570,000 SNPs. Samples that failed to reach 98.5% call rate were excluded from the study[35].

Individuals were only included if they had >97% European ancestry, as determined through an analysis of local ancestry[36]. Briefly, this analysis first partitions phased genomic data into short windows of ~100 SNPs. Within each window, a support vector machine is used to classify individual haplotypes into one of 31 reference populations. The support vector machine classifications are then fed into a hidden Markov model (HMM) that accounts for switch errors and incorrect assignments, and gives probabilities for each reference population in each window. Finally, simulated admixed individuals are used to recalibrate the HMM probabilities so that the reported assignments are consistent with the simulated admixture proportions. The reference population data are derived from public datasets (the Human Genome Diversity Project, HapMap and 1000 Genomes) and from 23andMe research participants who have reported having four grandparents from the same country.

A maximal set of unrelated individuals was chosen for each analysis using a segmental identity-by-descent (IBD) estimation algorithm[37]. Individuals were defined as related if they shared more than 700 cM IBD, including regions where the two individuals share either one or both genomic segments identical-by-descent. This level of relatedness (roughly 20% of the genome) corresponds approximately to the minimal expected sharing between first cousins in an outbred population. For the purposes of GWAS, if a skin cancer case was found to be related to a skin cancer control, the case was preferentially kept in the sample.

Participant genotype data were imputed against the March 2012 release of 1000 Genomes project reference haplotypes[38]. Data for each genotyping platform were phased and imputed separately. Variants that were only genotyped on the "V1" platform were flagged due to small sample size, and variants on chrM or chrY, because many of these are not currently called reliably. Using trio data, variants that failed a test for parent–offspring transmission were also flagged; specifically, the child's allele count was regressed against the mean parental allele count and variants with fitted $\beta < 0.6$ and $P < 10^{-20}$ for a test of $\beta < 1$ were flagged. Variants with a Hardy–Weinberg $P < 10^{-20}$ in Europeans, or a call rate of <90%, were also flagged. Genotyped variants were also tested for batch effects and variants with $P < 10^{-50}$ by analysis of variance of genotypes against a factor dividing genotyping date into 20 roughly equal-sized buckets were flagged. For imputed GWAS results, variants with average $r^2 < 0.5$ or minimum $r^2 < 0.3$ in any imputation batch were flagged, as well as SNPs that had strong evidence of an imputation batch effect, using an analysis of variance of the imputed dosages against a factor representing imputation batch; results with $P < 10^{-50}$ were flagged. Each variant flagged by QC on genotyped or imputation data were excluded from the GWAS analysis.

**Training and validation sets.** We split the full dataset in two parts: the first 103,008 participants collected between May and September 2016 were used as a training set; The 88,924 participants collected between October 2016 and June 2017 were used as a validation set (Supplementary Table 1). We selected samples to be between 30 and 90 years old and of European ancestry, and excluded <200 participants who reported outlier or inconsistent responses. The proportion of missing responses varied across questions from 2 to 15%. To obtain a complete dataset, we imputed missing phenotype data using *Hmisc* library in R (mean imputation using additive regression, bootstrapping, and predictive mean matching). To ensure robustness of inference, we duplicated our analyses using the non-imputed dataset, and compared the models. Without phenotype imputation, the number of participants dropped from 103,008 to 74,703 in the training set (Supplementary Table 5). In order to evaluate potential non-linear effect of age and complex relationship with the other risk factors, we also built and analyzed an age-matched dataset. We used semi-parametric and non-parametric matching methods implemented in MatchIt library in R, with a ratio of one case for two controls (or three controls for melanoma). The match dataset for the training set contained 14,672/29,288, 7307/14,513, and 3933/11,739 cases and controls for BCC, SCC, and melanoma, respectively (Supplementary Table 6).

**Prospective cohort and follow-up data.** Although every participant in the study received a follow-up survey every year after their initial completion, we only included participants from the validation set in the prospective cohort. Among the 88,924 participants from the validation set, 49,501 participants completed to at least one of the yearly follow-up surveys. For these participants, we computed the

number of incident cancers (controls at baseline and reporting to have been diagnosed in follow-up survey), and the number of skin cancer free participants (cases at baseline and not reporting skin cancer treatment in the follow-up survey).

**GWAS and polygenic risk score (PRS).** A genome-wide association study (GWAS) was conducted independently for BCC, SCC, and melanoma using data from the training set. Association with the phenotype was performed using logistic regression, including age, sex, the first five principal components, and variables representing the genotyping platform as covariates. Principal components were calculated using ~65,000 high quality genotyped and trans-ethnic variants that are present on all four genotyping platforms. GWAS analyses were run independently for the genotyped and imputed dosages. About 13M variants passed the pre- and post- GWAS QCs. The genomic control inflation factor was estimated as 1.06, 1.02, and 1.01 for BCC, SCC, and melanoma GWAS, respectively. PRS for each skin cancer was computed for all study participants (training and validating sets) using RiskPipe (v0.9; 23andMe Inc.)[39]. This pipeline uses a clumping and thresholding method to select variants ($P$-value threshold cut-off $<1.0e^{-6}$). The BCC, SCC, and melanoma PRS included 47, 14, and 18 variants, respectively (Supplementary Fig. 5). The statistical power to detect an additive association at a significance level of $1.0e^{-6}$ for a variant with a MAF = 0.1 and an OR = 1.2 was 0.95, 0.86, and 0.56 for BCC, SCC, and melanoma, respectively. PRS were calculated as the sum of the variant dosages weighted by their effect sizes, estimated in GWAS (Supplementary Table 10).

**Risk model development and selection.** Our construction of risk models for skin cancer consisted of three stages.

First, for each of the three skin cancers, independently, we first trained a predictive model including demographic factors (sex, age, and the same five principal components used in GWAS analysis), and published risk factors, present in the cancer survey (Supplementary Table 2). All predictive models presented in this study used a general linear model (GLM) with binomial distribution and logit link. All the analyses were performed in R (v3.2.5). After $P$-value selection ($<0.05$), the best models contained a total of 20 factors, the three demographic factors (after combining the 5 PCs into a unique ancestry risk factor) and 17 risk factors (Supplementary Table 4 and Supplementary Fig. 2a). Continuous variables, such as age were modeled with polynomial functions (2–4 degrees). For simplicity, and because the three skin cancers shared the majority of the risk factors, we decided to include the same set of risk factors for all three skin cancers: a factor was included if it passed the $P$-value selection in at least one of the three cancers. We also analyzed this 20-factor model in the non-imputed (Supplementary Table 5 and Supplementary Fig. 3a) and age-matched (Supplementary Table 6 and Supplementary Fig. 3b) datasets, to verify that the risk factor selection and effects were not biased by the phenotype imputation or some non-trivial effects of age.

Second, we explored the 23andMe phenotype database for unidentified risk factors. We curated 608 additional phenotypes with a missing rate not larger than 50% in the training set (Supplementary Table 3). We then fitted 608 GLMs, combining the 20-factor best model and each additional phenotype individually. We selected new risk factors with a conservative $P$-value $< 1e^{-8}$ criteria, and clustered them using ClustOfVar library in R, to identify correlated risk factors. The aggregation criterion is the decrease in homogeneity for the cluster being merged. The homogeneity of a cluster is the sum of the correlation ratio (for qualitative variables) and the squared correlation (for quantitative variables) between the variables and the center of the cluster which is the first principal component of PCAmix. PCAmix is defined for a mixture of qualitative and quantitative variables and includes ordinary principal component analysis (PCA) and multiple correspondence analysis (MCA) as special cases. A total of 67 phenotypes passed the $P$-value threshold in at least one of the three skin cancers, and were organized into 16 clusters (Supplementary Fig. 4). After discarding clusters related to general health or cancer treatments, we selected each 14 clusters the best factor, based on low $P$-value and low missing proportion. We then imputed the missing phenotypes (as previously described), fitted a unique GLM for each skin cancer, including the 20-factor model and the 14 newly discovered risk factors, and identified the best model with a $P$-value selection. The new models contained 31 risk factors (Supplementary Table 8 and Supplementary Fig. 2b). We computed the correlation matrix between these 31 risk factors (Supplementary Fig. 16).

Third, we added PRS to obtain the final full predictive models, including 32 risk factors (Supplementary Table 9 and Fig. 2).

As multicollinearity can be an issue for GLM analysis, we explored the potential for issues using mctest library in R. The Farrar-Glauber test detected significant multicollinearity in the three final 32-factor models, involving almost all factors included in the models. However, their variance inflation factors (VIF) were <2.2, at the exception of BMI and weight ($6.9 < VIF < 8.1$). We, therefore, decided to keep all the 32 factors in the final models because VIF > 10 is often used as a criteria[40] for excluding predictors, and we observed a good stability of the coefficient estimates in the different models built in the training and validation sets.

**Grouping risk predictors into risk scores.** We defined six risk scores by grouping risk factors included within the final 32-factor models (Figs. 1, 2). The

'Demographic risk score' includes three factors (age, sex, and ancestry, which is by itself a combination of the first five principal components). The 'Family history risk score' contains only one factor: it is defined as a simple score ranging from 0 to 4, where a value of 4 indicates that the participant reported that his/her father (+1), mother (+1), at least one sibling (+1), and at least one children (+1), developed skin cancer (BCC, SCC, or melanoma). We explored alternative and more complex definitions, including scores weighted by number of siblings, number of children, or skin cancer-specific family history. However, the simple score out performed these more complex scores at explaining phenotypic variance in the training set. As very few participants reported a score of 4, we combined the scores 3 and 4. The 'Mole risk score' combines four risk factors related to the presence or frequency of moles (dysplastic moles, presence of large moles, number of moles on the right arm), and skin conditions (diagnosed with actinic keratosis before the age of 40). The 'Susceptibility risk score' combines 8 factors related to pigmentation but also skin reaction to sun exposure (skin, eye, and hair colors, number of freckles on face and body, number of blisters caused by sunburns, and sun hair lightening). The 'Exposure risk score' combines 8 factors that estimate lifetime or current weekly sun exposure (sunbathing frequency before age 30, tanning bed usage, childhood and adulthood latitude and elevation, typical sun exposure per week, outdoor job, and physical activity). The 'Miscellaneous risk score' combines 7 factors that are not a natural fit in the 5 other risk scores. These risk factors are mainly related to metabolism and personality/behavior (BMI, weight, smoking, alcohol consumption, seasonal allergies, being a 'morning person', and preference for keeping a 'clean desk').

We finally constructed two main disease risk scores, called DRS and DRSA. DRS includes all the factors from 32-factor model. DRSA excludes the effects of age. Similar to the PRS computation, the risk scores are calculated as the sum of the factor responses weighted by their effect sizes, estimated by the 32-factor model in the training set (Supplementary Table 14). Risk scores were computed for each participant included in the training and validation sets.

**Risk scores correlation and age dependency.** We explored the relationship of the risk scores in the validation set. We first calculated pairwise Spearman correlation coefficients of the different risk scores within and between skin cancer types in the validation set (Supplementary Table 11). As expected, PRS were independent of the participant age at baseline. We also showed that the DRSA were also mostly uncorrelated with the participant age at baseline: while the DRSA averages showed a slight increase around age 50 for the three skin cancers, the variances around these estimates are very large, and the age dependence of DRSA is minimal (Supplementary Fig. 8). We finally explored the relationship between the PRS, Exposure, Susceptibility, Mole, Family, and Miscellaneous risk scores included in the DRSA by running principal component analyses (*prcomp* library in R, without scaling). The three skin cancers showed different risk score signatures, in particular for the contribution of Exposure, Susceptibility, and Family risk scores. The PRS and Mole risk scores have the largest and generally orthogonal effects on the first two PCs (Supplementary Fig. 9).

We also explored the potential age dependency of DRSA (Supplementary Fig. 16) but also of individual risk scores (Supplementary Fig. 19), for participants with low, middle, and high risk of developing skin cancer: all the risk scores showed minimal age dependency within the different risk groups.

**Risk score prediction validation.** For all the risk scores, we computed the receiver operating characteristic (ROC) curve in both the training and validating sets, using ROCR library in R, and we extracted the area under the curve (AUC) values. We also computed the precision-recall (PR) curves (Supplementary Fig. 6).

**Risk score distribution tails and clinical features of skin cancer cases.** In the validation set, we binned the risk score distributions by percentiles (~890 participants per percentile), and evaluated different attributes of the three skin cancers, using the baseline survey data. We first calculated the prevalence per bin of participants who reported skin cancer. All risk scores showed an increase of skin cancer with higher risk score values. They also showed long tails, as observed in PRS studies (Fig. 3 and Supplementary Fig. 7), with strong enrichment of participants reporting skin cancer in percentiles >95%. For the participants reporting skin cancer at baseline, we computed the mean age at diagnosis (Fig. 4 and Supplementary Fig. 10), the mean cancer stage (I–IV) at diagnosis, the mean number of diagnosis/treatments for a specific skin cancer during the previous 2 years, and mean number of total skin cancer diagnosis (from 1 for only one of the three skin cancer to 3, if diagnosed for BCC, SCC, and melanoma) per percentiles (Table 1).

Using the prospective data, we computed the number and proportion of incident cases (the proportion of participants who did not report, at baseline, to have been diagnosed with skin cancer, and in the follow-up surveys, reported to be diagnosed for skin cancer during the previous year). The number and proportion of cancer free participants (the proportion of participants who reported, at baseline, to have been diagnosed with skin cancer, and in the follow-up survey, not been treated for skin cancer during the previous year) in each percentile bin (Table 1, Supplementary Figs. 11, 12).

**Interaction analysis**. We explored more complex prediction skin cancer models by including and testing pairwise interaction. We first tested for all pairwise interaction between the 7 risk factors using GLMs (Supplementary Table 12 and Supplementary Fig. 13). Because the large effect of age, we tested independently the interaction terms with age, sex, and ancestry from the Demographic risk score. We also tested pairwise interactions between the top factors of each risk score (Supplementary Table 13). Although we identified several significant pairwise interaction, they were explaining only a small amount of the total variance (Supplementary Fig. 13), and only improved marginally skin cancer prediction (results not shown).

**Incidence rates**. We computed the incidence rates ($I$) from the prevalence ($P$) in the validation set. For $m$-year incidence rate at age $a$:

$$I_a = \frac{P_a - P_{a-m}}{1 - P_{a-m}}$$

**Observed and expected age of diagnosis**. We compared the observed and expected ages of diagnosis in the validation set. The expected ages of diagnosis in Fig. 5 were computed for each DRS-DL curve, by deriving the yearly incidence rates ($I$) from the lifetime risks associated to each curve. To model the age distribution in the US population, we extracted the estimated yearly survival rates ($S$) produced by the CDC for white Americans (sex combined)[41]. The expected mean and SD of age of diagnosis ($m$) was obtained with:

$$m = \frac{1}{\sum_{a=30}^{90} S_a I_a} \sum_{a=30}^{90} S_a I_a a \text{ and } SD = \sqrt{\frac{\sum_{a=30}^{90} S_a I_a (a-m)^2}{\sum_{a=30}^{90} S_a I_a}}.$$

**Reporting summary**. Further information on research design is available in the Nature Research Reporting Summary linked to this article.

## Data availability

Full summary statistics for the three skin cancer GWAS will be made available to qualified researchers under an agreement that protects participant privacy. Researchers should visit https://research.23andme.com/dataset-access/ for more details and instructions for applying for access to the data. The remaining data are available within the Article, Supplementary Information or available from the authors upon request.

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

## Acknowledgements

We thank the research participants and employees of 23andMe for making this work possible.

## Author contributions

P.F., B.A., S.J.P., R.G., and A.A. designed this study. M.J. and C.H.W. developed the survey. The 23andMe Research Team acquired and processed the data. N.A.F. implemented PRS pipeline. P.F. analyzed the data. P.F., B.A., S.J.P., R.G., and A.A. interpreted the data and P.F. wrote the first draft of the manuscript. All authors participated in the preparation of the manuscript by reading and commenting on drafts prior to submission.

## Competing interests

All authors are current or former employees of 23andMe, Inc., and hold stock or stock options in 23andMe.

## Additional information

## 23andMe Research Team

Michelle Agee[1], Robert K. Bell[1], Katarzyna Bryc[1], Sarah L. Elson[1], David A. Hinds[1], Karen E. Huber[1], Aaron Kleinman[1], Nadia K. Litterman[1], Jennifer C. McCreight[1], Matthew H. McIntyre[1], Joanna L. Mountain[1], Elizabeth S. Noblin[1], Carrie A. M. Northover[1], J. Fah Sathirapongsasuti[1], Olga V. Sazonova[1], Janie F. Shelton[1], Suyash Shringarpure[1], Chao Tian[1], Joyce Y. Tung[1] & Vladimir Vacic[1]

