## [Peer Review File · Nature Communications]

REVIEWERS' COMMENTS

Reviewer #1 (Remarks to the Author):

The authors did an excellent job of addressing all of my major concerns. There are only two minor points that came up in reading the revised version of the manuscript.

1. It is interesting that although some of the same genes are coming up in the skin type dependent models, the variants associated with those genes vary (e.g. HERC2 and MC1R for SCC and melanoma). Could the authors comment on why this is? Can the variants be used interchangeably in the disease specific PRS?

2. On page 5 the authors use AUC and on page 7 (in the methods) they spell out area under the curve (AUC). The first time using AUC should be spelled out and then just AUC can be used in the methods.

Reviewer #4 (Remarks to the Author):

The manuscript is now suitable for publication in the journal.

Reviewer #5 (Remarks to the Author):

As I have been asked to review the response to reviewer 3 I have indicated where comments/quotes from reviewer 3 (R3), authors (A), or myself (Ex). In general they have adequately addressed concerns. There are only a few minor points where they have incompletely addressed concerns – I have noted these with (Action)

Review of response to reviewer #3

Reviewer #3 (Remarks to the Author):

R3) - The outcomes were self-reported. Multiple studies have shown that melanoma is poorly self-reported, partly due to confusion between different types of skin cancer

(A) See responses to Reviewer 1, questions C.1 and F.1. We added a paragraph in the discussion specifically tacking self-reported data and misclassification.

(Ex) They have sufficiently addressed this concern

(R3) - It was only a short period of follow-up (this is more relevant for non-genetic factors)

(A) The main goal of a prospective cohort is identifying incident cases. The follow-up length is generally determined by aiming at a duration that would give sufficient....

(Ex) They have sufficiently addressed this concern

(R3) There was low participation rate in the follow-up survey, which could lead to selection bias (49,501/88,924)

(A) We did not see evidence of biases in the prospective cohort relative to the baseline cohort (Table S1).

(Ex) While they technically address this issue – the prospective component of the test cohort is similar to the baseline component (Table 1) the validation set does have higher rates of skin cancer overall versus the training set; however I am not sure this is a major concern.

(R3) - External validation in a different dataset altogether would strengthen the paper, although limited relevant datasets are available.

(A) We agree with the reviewer. Unfortunately, we are not aware of a dataset, with all the required genetics and non-genetics information, that would permit to replicate our predictive models. All the factor weights required to replicate our findings are given in the Supplementary Material.

(Ex) They have sufficiently addressed this concern

(R3) - The Abstract states that “High DRS were associated with an up to 13-fold increase in the risk of developing skin cancer” but this is a bit misleading for participants as it only refers to the top percentile compared to the average risk score. It would be more appropriate to show the odds ratio per standard deviation difference, or for the top quartile. For whatever is reported, it needs to be clear in the text what is being compared.

(A) We added the odds ratio per standard deviation increase in score to facilitate the comparison with results from published studies using smaller cohorts (under 'Risk score prediction performance and correlations'). However, odds ratio per standard deviation increase (or also interquartile increase) are only rough estimates of OR change across the risk score distribution. With large sample sizes permitting to evaluate parameters per percentile of risk score distribution, we believe that Fig. 3 is a better representation of the dynamic of skin cancer risk across risk score distributions and in particular for visualizing the dynamic in the tails of the distributions.

(Ex) I do not think this have adequately addresses this concern.

(Action) The authors either need to be explicit in the abstract what they mean by high DRS (top 1% vs middle) and/or report the change per SD (including 95% CI as the per following comment). I personally think both would be useful. Likewise when discussing figure S11, S12 it needs to be clear that observed earlier development of skin cancer results from comparing the top 1% versus the middle; the comparison for example for the top 20% would be less dramatic.

(R3) Although the three skin cancers share many common risk factors, they are not always identical and displays some differences in etiology and strength of association (e.g. Krickler et al, *Photochem Photobiol.* 2017 Nov;93(6):1483-1491. doi: 10.1111/php.12807). Perhaps separate models should be considered for each cancer rather than combining into one 32-factor model?

(A) It seems that the reviewer understood that we built a unique model but we did in fact build one for each skin cancer: On page 4, "The final 32-factor models were obtained independently for each skin cancer, following the procedure described in Fig. 1c (see Methods for a detailed description). For simplicity and because the three skin cancers shared common risk factors, the 32-factor models combined all risk factors identified for each skin cancer." We changed 'combined' to 'included' to mitigate the possible confusion.

(Ex) While the authors have not (technically) adequately addressed this concern – they have developed 3 separate models, and could test them – I am not convinced separate models would improve the manuscript, so am satisfied they have addressed this concern in principle. The inclusion of risk factors associated with (for example) one cancer but not the other two would not adversely impact the models, and is simpler.

(Action) I would like to note though that this section of the methods describing how the models were built is fairly hard to follow and may benefit from a rewrite.

(R3) In the Results section and abstract, it would be useful to report confidence intervals for the major findings

(A) The two metrics reported in the abstract ("up to 13-fold increase" and "by up to 14 years") have no "natural" errors: they are simply comparing the means within two percentiles. We could force to have a 95% CI, by jackknifing for example, but the mean estimates are based on a very large number of participants, the SE is extremely small and the 95% CI is quasi indistinguishable from the mean. The bulk of the analysis results, including P-values and confidence intervals are shown in the Supplementary Material (in particular Table S9 for the full statistical description of the three final models). Figure 2 is a simplified visualization of the complete results and showed examples of factor

behaviors, including mean and 95% CI. Fig 3 and 4 are showing the full dynamic of the skin cancer risk and age of onset across the distribution of risk scores. Fig 4 includes a 95% CI for age of onset.

(Ex) Barring above – where I agree a value requiring a CI (the OR per SD) should be in the abstract – this has been addressed.

(R3) BMI and weight are highly correlated and it is usually not appropriate to put both variables in a model together. I suggest adding weight and height as separate variables

instead.

(A) The reviewer is correct; BMI and weight are highly correlated. However, it is far from a complete correlation, and they are potentially capturing differently social and behavioral information associated with being under- or over-weight (see the discussion in page 4). If the reviewer is concerned by collinearity.....

(Ex) Adequately addressed

(R3) Actinic keratosis is not related to moles so suggest removing it from the 'Mole risk score'

(A) We clarified page 6 that 'Mole risk score' included mole and skin damage factors.

(Ex/Action) While a minor concern I agree with the reviewer – this variable should be called something other than just Mole if it includes something other than just moles. Mole-AK?

(R3) What was the rationale for creating a score without demographic factors?

(a) To build a score independent of age. We were interested on understanding the disease trajectory of high risk individuals and the dynamic of the age of onset....

(Ex) Adequately addressed

(R3) Why did you choose a clumping and thresholding method for selecting the genetic variants for the risk score? Why not choose published variants?

(A) See response to Reviewer 1 question F1.

(Ex) Adequately addressed

(R3) On page 7, it states “About 41.1%, 20.0%, and 9.9% of the participants with PRS in the top percentile reported developing BCC, SCC, and melanoma respectively...” but does not give a timeframe for these percentages – is this over a two-year follow-up period?

(A) It is in the validation set, at baseline. They are not incidence estimates but the proportion of participants that had skin cancer before the baseline survey.

(Ex/Action) While adequately addressed but the authors should modify page 7 so this is clearer

R3) It is mentioned that “all DRS were heavily driven by current age, and they cannot be used directly in early detection programs.” Why does this preclude using this score for early detection, if age is important? The authors state (p6/7) “It suggests that the different risk factors included in DRS-DL estimated lifetime risks of skin cancer, and could be collected across adult life (post 30 years old).” Can you expand on this rationale to be clarify your argument.

(A) We expanded the discussion to about the usage and applications of the two disease risk scores.

(Ex) Adequately addressed

(R3) How was cancer recurrence measured in the questionnaire?

(A) We added a complete description in the main text (in addition to the one present in Methods) in page 8:

(Ex) Adequately addressed

(R3) How did the authors deal with correlation between different risk factors including genetic and non-genetic risk factors?

(A) See the response above, to the BMI and weight question.

(Ex) Adequately addressed

REVIEWERS' COMMENTS

Reviewer #1 (Remarks to the Author):

The authors did an excellent job of addressing all of my major concerns. There are only two minor points that came up in reading the revised version of the manuscript.

1. It is interesting that although some of the same genes are coming up in the skin type dependent models, the variants associated with those genes vary (e.g. HERC2 and MC1R for SCC and melanoma). Could the authors comment on why this is? Can the variants be used interchangeably in the disease specific PRS?

The skin cancers share a large part of their respective genetic architecture, in particular the regions involve in pigmentation. However, the effect estimates for these variants are not interchangeable between the three skin cancers (same variants or haplotypes but difference contributions to specific skin cancers).

2. On page 5 the authors use AUC and on page 7 (in the methods) they spell out area under the curve (AUC). The first time using AUC should be spelled out and then just AUC can be used in the methods.

We followed the reviewer's recommendation, and defined AUC in page 4, and then used the abbreviation in the rest of the manuscript.

Reviewer #4 (Remarks to the Author):

The manuscript is now suitable for publication in the journal.

Reviewer #5 (Remarks to the Author):

As I have been asked to review the response to reviewer 3 I have indicated where comments/quotes from reviewer 3 (R3), authors (A), or myself (Ex). In general they have adequately addressed concerns. There are only a few minor points where they have incompletely addressed concerns – I have noted these with (Action)

Review of response to reviewer #3

Reviewer #3 (Remarks to the Author):

(R3) - The outcomes were self-reported. Multiple studies have shown that melanoma is poorly self-reported, partly due to confusion between different types of skin cancer

(A) See responses to Reviewer 1, questions C.1 and F.1. We added a paragraph in the discussion specifically tacking self-reported data and misclassification.

(Ex) They have sufficiently addressed this concern

(R3) - It was only a short period of follow-up (this is more relevant for non-genetic factors)

(A) The main goal of a prospective cohort is identifying incident cases. The follow-up length is generally determined by aiming at a duration that would give sufficient....

(Ex) They have sufficiently addressed this concern

(R3) There was low participation rate in the follow-up survey, which could lead to selection bias (49,501/88,924)

(A) We did not see evidence of biases in the prospective cohort relative to the baseline cohort (Table S1).

(Ex) While they technically address this issue – the prospective component of the test cohort is similar to the baseline component (Table 1) the validation set does have higher rates of skin cancer overall versus the training set; however I am not sure this is a major concern.

(R3) - External validation in a different dataset altogether would strengthen the paper, although limited relevant datasets are available.

(A) We agree with the reviewer. Unfortunately, we are not aware of a dataset, with all the required genetics and non-genetics information, that would permit to replicate our predictive models. All the factor weights required to replicate our findings are given in the Supplementary Material.

(Ex) They have sufficiently addressed this concern

(R3) - The Abstract states that "High DRS were associated with an up to 13-fold increase in the risk of developing skin cancer" but this is a bit misleading for participants as it only refers to the top percentile compared to the average risk score. It would be more appropriate to show the odds ratio per standard deviation difference, or for the top quartile. For whatever is reported, it needs to be clear in the text what is being compared.

(A) We added the odds ratio per standard deviation increase in score to facilitate the comparison with results from published studies using smaller cohorts (under 'Risk score prediction performance and correlations'). However, odds ratio per standard deviation increase (or also interquartile increase) are only rough estimates of OR change across the risk score distribution. With large sample sizes permitting to evaluate parameters per percentile of risk score distribution, we believe that Fig. 3 is a better representation of the dynamic of skin cancer risk across risk score distributions and in particular for visualizing the dynamic in the tails of the distributions.

(Ex) I do not think this have adequately addresses this concern.

(Action) The authors either need to be explicit in the abstract what they mean by high DRS (top 1% vs middle) and/or report the change per SD (including 95% CI as the per following comment). I personally think both would be useful. Likewise when discussing figure S11, S12 it needs to be clear that observed earlier development of skin cancer results from comparing the top 1% versus the middle; the comparison for example for the top 20% would be less dramatic.

We updated the abstract and followed the reviewer recommendation of presenting the fold change and the OR. Unfortunately, because the abstract has a limited size, we could report the OR (and 95% CI) for the three skin cancers. We decided to report here the minimal OR across the three skin cancer (but the details, including 95%CI is present in the results section.

" Top percentile DRS was associated with an up to 13-fold increase (odds ratio per standard deviation increase > 2.5) in the risk of developing skin cancer relative to the middle DRS percentile."

We also clarified the fold change in page 4:

"These correspond to a 5.2x, 8.1x, and 12.9x risk increase relative to participants with middle DRS percentile."

(R3) Although the three skin cancers share many common risk factors, they are not always identical and displays some differences in etiology and strength of association (e.g. Krickler et al, Photochem Photobiol. 2017 Nov;93(6):1483-1491. doi: 10.1111/php.12807). Perhaps separate models should be considered for each cancer rather than combining into one 32-factor model?

(A) It seems that the reviewer understood that we built a unique model but we did in fact build one for each skin cancer: On page 4, "The final 32-factor models were obtained independently for each skin cancer, following the procedure described in Fig. 1c (see Methods for a detailed description). For simplicity and because the three skin cancers shared common risk factors, the 32-factor models combined all risk factors identified for each skin cancer." We changed 'combined' to 'included' to mitigate the possible confusion.

(Ex) While the authors have not (technically) adequately addressed this concern – they have developed 3 separate models, and could test them – I am not convinced separate models would improve the manuscript, so am satisfied they have addressed this concern in principle. The inclusion of risk factors associated with (for example) one cancer but not the other two would not adversely impact the models, and is simpler.

(Action) I would like to note though that this section of the methods describing how the models were built is fairly hard to follow and may benefit from a rewrite.

The reviewer did not give specific recommendations and it was not clear which parts were hard to follow. Because the analysis had multiple dependent steps, we knew that it could require additional efforts from the readers: we created the analysis flow chart in Figure 1 to clarify and simplify the analysis process.

(R3) In the Results section and abstract, it would be useful to report confidence intervals for the major findings

(A) The two metrics reported in the abstract ("up to 13-fold increase" and "by up to 14 years") have no "natural" errors: they are simply comparing the means within two percentiles. We could force to have a 95% CI, by jackknifing for example, but the mean estimates are based on a very large number of participants, the SE is extremely small and the 95% CI is quasi indistinguishable from the mean. The bulk of the analysis results, including P-values and confidence intervals are shown in the Supplementary Material (in particular Table S9 for the full statistical description of the three final models). Figure 2 is a simplified visualization of the complete results and showed examples of factor behaviors, including mean and 95% CI. Fig 3 and 4 are showing the full dynamic of the skin cancer risk and age of onset across the distribution of risk scores. Fig 4 includes a 95% CI for age of onset.

(Ex) Barring above – where I agree a value requiring a CI (the OR per SD) should be in the abstract – this has been addressed.

(R3) BMI and weight are highly correlated and it is usually not appropriate to put both variables in a model together. I suggest adding weight and height as separate variables instead.

(A) The reviewer is correct; BMI and weight are highly correlated. However, it is far from a complete correlation, and they are potentially capturing differently social and behavioral information associated with being under- or over-weight (see the discussion in page 4). If the reviewer is concerned by collinearity.....

(Ex) Adequately addressed

(R3) Actinic keratosis is not related to moles so suggest removing it from the 'Mole risk score'

(A) We clarified page 6 that 'Mole risk score' included mole and skin damage factors.

(Ex/Action) While a minor concern I agree with the reviewer – this variable should be called something other than just Mole if it includes something other than just moles. Mole-AK?

The five environmental risk scores are all heterogeneous and combine different types of risk factors. It is true for Mole but also for Exposure. We tried to label them to represent the factor(s) with the largest contribution. The Mole risk score contains 3 mole risk factors (with large effects) and AK.

(R3) What was the rationale for creating a score without demographic factors?

(a) To build a score independent of age. We were interested on understanding the disease trajectory of high risk individuals and the dynamic of the age of onset....

(Ex) Adequately addressed

(R3) Why did you choose a clumping and thresholding method for selecting the genetic variants for the risk score? Why not choose published variants?

(A) See response to Reviewer 1 question F1.

(Ex) Adequately addressed

(R3) On page 7, it states "About 41.1%, 20.0%, and 9.9% of the participants with PRS in the top percentile reported developing BCC, SCC, and melanoma respectively..." but does not give a timeframe for these percentages – is this over a two-year follow-up period?

(A) It is in the validation set, at baseline. They are not incidence estimates but the proportion of participants that had skin cancer before the baseline survey.

(Ex/Action) While adequately addressed but the authors should modify page 7 so this is clearer

We clarified it in page 7:

" the prevalence of skin cancer for participants in the top DRS percentile was 69.8%, 46.7%, and 31.8% for BCC, SCC, and melanoma, respectively."

(R3) It is mentioned that "all DRS were heavily driven by current age, and they cannot be used directly in early detection programs." Why does this preclude using this score for early detection, if age is important? The authors state (p6/7) "It suggests that the different risk factors included in DRS-DL estimated lifetime risks of skin cancer, and could be collected across adult life (post 30 years old)." Can you expand on this rationale to be clarify your argument.

(A) We expanded the discussion to about the usage and applications of the two disease risk scores.

(Ex) Adequately addressed

(R3) How was cancer recurrence measured in the questionnaire?

(A) We added a complete description in the main text (in addition to the one present in Methods) in page 8:

(Ex) Adequately addressed

(R3) How did the authors deal with correlation between different risk factors including genetic and non-genetic risk factors?

(A) See the response above, to the BMI and weight question.

(Ex) Adequately addressed